# Structural Study and Detrital Zircon Provenance Analysis of the Cycladic Blueschist Unit Rocks from Iraklia Island: From the Paleozoic Basement Unroofing to the Cenozoic Exhumation

**Sofia Laskari** [1,*]**, Konstantinos Soukis** [1]**, Stylianos Lozios** [1]**, Daniel F. Stockli** [2]**, Eirini M. Poulaki** [2]
**and Christina Stouraiti** [1]

[1] Faculty of Geology and Geoenvironment, National and Kapodistrian University of Athens,
15784 Athens, Greece; soukis@geol.uoa.gr (K.S.); slozios@geol.uoa.gr (S.L.); chstouraiti@geol.uoa.gr (C.S.)

[2] Department of Geological Sciences, Jackson School of Geosciences, University of Texas at Austin,
Austin, TX 78712, USA; stockli@jsg.utexas.edu (D.F.S.); eirini_poulaki@utexas.edu (E.M.P.)

[*] Correspondence: slaskari@geol.uoa.gr

**Abstract:** Detailed mapping and structural observations on the Cycladic Blueschist Unit (CBU) on Iraklia Island integrated with detrital zircon (DZ) U-Pb ages elucidate the Mesozoic pre-subduction and the Cenozoic orogenic evolution. Iraklia tectonostratigraphy includes a heterogeneous Lower Schist Fm., juxtaposed against a Marble Fm. and an overlying Upper Schist Fm. The contact is an extensional ductile-to-brittle-ductile, top-to-N shear zone, kinematically associated with the Oligo-Miocene exhumation. The DZ spectra of the Lower Schist have Gondwanan/peri-Gondwanan provenance signatures and point to Late Triassic Maximum Depositional Ages (MDAs). A quartz-rich schist lens yielded Precambrian DZ ages exclusively and is interpreted as part of the pre-Variscan metasedimentary Cycladic Basement, equivalent to schists of the Ios Island core. The Upper Schist represents a distinctly different stratigraphic package with late Cretaceous MDAs and dominance of Late Paleozoic DZ ages, suggestive of a more internal Pelagonian source. The contrast in the DZ U-Pb record between Lower and Upper Schist likely reflects the difference between a Paleotethyan and Neotethyan geodynamic imprint. The Triassic DZ input from eroded volcanic material is related to the final Paleotethys closure and Pindos/CBU rift basin opening, while late Cretaceous metamorphic/magmatic zircons and ~48–56 Ma zircon rims constrain the onset of Neotethyan convergence and high-pressure subduction metamorphism.

**Keywords:** Cycladic Blueschist Unit; detrital zircon U-Pb dating; Iraklia Island; Maximum Depositional Age; provenance; Paleotethys; Paros-Naxos Detachment System; ductile shear zone

## 1. Introduction

In most cases, poly-deformed and poly-metamorphosed high-grade rocks have lost their fossil record and much of the stratigraphic framework necessary for local and regional correlations. In ancient subduction complexes, this lack of a stratigraphic frame hinders the reconstruction of subduction and exhumation processes, hampers an understanding of these processes' overprint primary structures and lithostratigraphy and leads to the resulting tectonostratigraphy due to both subduction underplating and back-arc extension.

The recognition of Metamorphic Core Complexes (MCCs) [1] made the Attic-Cycladic Crystalline Complex (ACCC) (Figure 1a,b) the subject for several studies [2–15] primarily focused on the Cenozoic structural and metamorphic evolution of the overlying Permian to Mesozoic Cycladic Blueschist Unit (CBU) and its Carboniferous Cycladic Basement (CB). These studies have offered insights into both the subduction processes and partly the back-arc extension and MCC-formation processes [2–15]. However, despite a rare accumulation of constraints, several fundamental aspects of pre-orogenic configuration are

still not fully understood. One of them is how the transition from the Paleozoic Paleotethys subduction cycle to Neotethys opening is imprinted into the CBU rock record.

The small island of Iraklia occupies a critical position between Ios and Naxos (Figure 1c), two islands where pre-Alpine CB and the CBU rocks constitute the footwall of MCCs exhumed by sets of extensional detachments, including the Paros–Naxos detachment system. Although previous studies [16,17] reported that Iraklia comprises CBU rocks mainly similar to those of Naxos and Ios, the authors of [18] suggested the existence of pre-alpine basement rocks. However, no detailed work has been carried out to investigate (a) the tectonostratigraphy of Iraklia, (b) the possible stratigraphic impact of the Paleotethyan and Neotethyan plate-tectonic cycles, and (c) the stratigraphic/metamorphic correlations with rocks in the footwalls of the Ios and Naxos MCCs.

Recently, detrital zircon (DZ) U-Pb studies in the ACCC have provided fundamental insights regarding the Maximum Depositional Ages (MDAs), tectonic affinity and provenance of the CBU and CB rocks, and the age of metamorphism. A number of these studies have also shed light on the CBU stratigraphy and the pre-Eocene subduction paleogeographic organization [14,15,19–27].

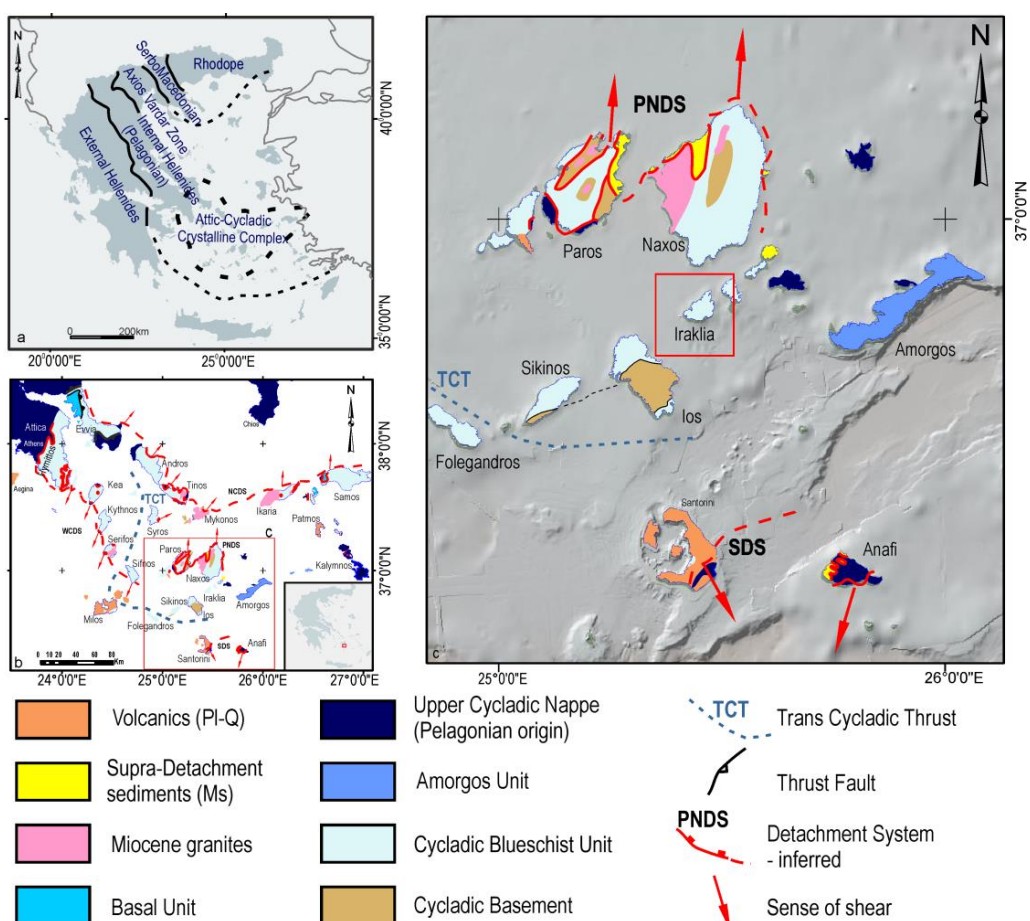

**Figure 1.** (**a**) Schematic map of major structures in Greece, (**b**) detailed map of the ACCC illustrating the main units, the major structures, and (**c**) simplified geology of Iraklia and neighboring islands (base map made from GeoMapApp https://www.geomapapp.org, accessed on 20 December 2021). Trans Cycladic Thrust (TCT), North Cycladic Detachment System (NCDS), Paros-Naxos Detachment System (P-NDS) West Cycladic Detachment System (WCDS), and Santorini Detachment System (SDS) compiled from [3,5,7–10,13,26–30].

This study combines detailed geological mapping, structural observations, and DZ U-Pb geochronology to (a) constrain the tectonostratigraphy of Iraklia CBU rocks, (b) establish

tectonostratigraphic and metamorphic correlations with the neighboring Ios and Naxos, and (c) reconstruct the pre-subduction history of Iraklia, as part of the regional Paleotethyan to Neotethyan evolution.

## 2. Geological Setting

The ACCC exposes the deepest exhumed parts of the subducted and metamorphosed Hellenides as a result of Cenozoic back-arc extension due to accelerated slab roll-back of the subducting plate [6,8,31]. The tectonostratigraphy of the ACCC comprises the pre-alpine CB and the metasedimentary CBU cover sequence, which constitute the exhumed lower plate of regional low-angle detachment fault systems. The upper plate comprises the Upper Cycladic Nappe and the Late Miocene supra-detachment sediments [6,8,18,27,32–37].

The CBU includes a Permo-Mesozoic sequence of marble, schists, and metavolcanic rocks metamorphosed under eclogite to blueschist metamorphic conditions in the Eocene, which are best preserved on Evvia, Syros, Sifnos, and the Tinos islands [27,38–42]. While most classic CBU rocks experienced peak HP-LT metamorphism during the Eocene (~45–53 Ma) [43–45], recent studies have documented either prolonged or renewed early Eocene to Oligocene HP metamorphic event during ongoing subduction and underplating [46,47]. During the Oligo-Miocene, the CBU and CB rocks have been reworked at a much lower pressure in various temperature conditions from greenschists to amphibolite facies and locally experienced partial melting conditions due to rapid syn-extensional decompression and migration of the Hellenic magmatic arc (e.g., Naxos, Paros, Ikaria and Mykonos [35,48–55]).

An overall two-stage Cenozoic exhumation history has been documented in most CBU rocks. The earlier Eocene-Oligocene syn-orogenic exhumation stage occurred within the subduction zone, allowing for local preservation of the HP imprint (e.g., Evvia, Syros, Tinos, Sifnos) [27,41,46,56,57]. The second stage included a late Cenozoic (mainly Miocene) post-orogenic exhumation, resulting in the formation of the MCCs bounded by low-angle detachment systems [6–8,34,55–60]. The major detachment systems that have been identified juxtaposing the CBU against the Upper Cycladic Nappe are: (i) the North Cycladic Detachment System (Andros, Tinos, Mykonos, Ikaria) and the Paros-Naxos Detachment System (Paros, Naxos, Ios) with top-to-the-N-NE kinematics, and (ii) the West Cycladic Detachment System (Attica, Kea, Kythnos, Serifos) and the South Cycladic Shear Zone and the Santorini Detachment (Ios, Santorini) with top-to-the-S-SW kinematics, supporting the scenario of a symmetric pattern of extension in the Aegean Sea [1,3,4,6–10,13,29,61–65]. Emplacement of syn-tectonic Miocene I- and S-type granitic intrusions due to migration of the magmatic appears to temporally coincide with the late-stage activity of these detachment systems [5,27,49,51,66–69].

The structurally underlying CB is mainly exposed on the islands of Paros, Naxos, Ios, Sikinos and comprises Carboniferous (Variscan s.l.) granitoids related to the closure of the Paleotethys and intruded into late Proterozoic–early Paleozoic metasedimentary sequences of Gondwanan affinity [14,19,20,32,35,70–77]. The contact between CB/CBU has been variably considered as either an extensional detachment fault, ductile shear zone [78–81], or thrust [58]. While undoubtedly tectonically overprinted in a number of places, recent detrital zircon U-Pb studies have documented a deformed depositional CBU-CB contact with the CBU resting unconformably on the CB and representing continuous sedimentation from the late Permian to late Cretaceous [15,23,25,82,83]. This suggests a parautochthonous nature for the contact between CB and CBU that is also supported by the similarity of DZ signatures of the basal CBU, closely resembling the underlying CB [15,83]. The DZ signatures of the CB and CBU metasedimentary units throughout the Cyclades revealed a Gondwanan provenance [20] that remained relatively unchanged until the CBU rocks approached the trench of the subduction zone, as indicated by significant late-stage input from the Internal Hellenides [14,15,25].

Structurally, above the CBU lies the Upper Cycladic Nappe. This is a diverse group of low-grade to amphibolite-facies Permian to Mesozoic metasedimentary rocks, granitic

orthogneiss, and metamorphosed ophiolitic rocks (the latter locally known as Upper Unit) [4,10,11,28,32,34,37,66,71,84,85]. The Upper Cycladic Nappe is intruded by sparse late Cretaceous granitoids, and it is correlated with the Pelagonian domain of continental Greece [4,11,28,32,34,71,84–86]. The Pelagonian-derived rocks of continental Greece exhibit a complete stratigraphy, consisting of Triassic to Jurassic carbonates underlying a wild-flysch and ophiolites of late Jurassic to early Cretaceous age, covered by late Cretaceous transgressive carbonates and a late Cretaceous to Eocene flysch [27,86].

The geology of Iraklia island was first studied by the Greek Geological survey, which published a 1:50,000 scale geological map for the Iraklia and Schinoussa islands [16]. These workers reported the existence of a probable Triassic marble sequence, overlying schist and marble alternations, which they assigned to CBU. Subsequent structural and piezometric paleo-stress analyses were used to propose a four-stage deformation that includes an early D1 deformation attributed to the Eocene metamorphism, D2 shearing with top-to-the-north kinematics [17]. The syn-tectonic temperature conditions were constrained to 400–450 °C and differential stress approximately 29–62 MPa. In contrast, D3 and D4 deformation phases were attributed to E-W shortening, which resulted in isoclinal folding and reverse faulting. The same authors reported blueschists facies assemblages from the southwestern part of the island, suggesting that the main deformation phase occurred within the stability field of glaucophane.

Although the lithostratigraphy of Iraklia is quite well known, age constraints, the pre-subduction evolution, and the tectonostratigraphic relationships with the neighboring islands, especially of Naxos and Ios, where both the CB and CBU crop out, are still pending. In the following paragraphs, a new map and revised lithostratigraphy will be presented, based on new constraints on the structural evolution, DZ U-Pb sedimentary provenance, tectonic affinity and age of the CBU rocks of Iraklia.

## 3. A New Geological Map of Iraklia

### 3.1. Lithostratigraphic Subdivision

Based on the new geological mapping results, the lithostratigraphic column of CBU on Iraklia Island can be divided into two packages, juxtaposed by a ductile to brittle-ductile shear zone. The lower part includes the Lower Schist Formation, which is exposed mainly at the central and southern part of the island and a limited outcrop in the north-eastern region (Figure 2). The Lower Schist Fm. is lithologically quite heterogeneous and includes light brown quartz-mica schists interlayered locally with isoclinally folded light gray to dark blue marble, paragneiss and quartz-rich schist lenses.

The upper part of the lithostratigraphic column includes a thick Marble Fm. and an overlying Upper Schist Fm. The Upper Schist Fm. comprises light green to light brown intensively quartz-chlorite mica-schist intercalated with thin marble layers (Figures 2 and 3a). The thick Marble Fm. occupies the structural position between the Lower and Upper Schist Fm. and is a well-foliated calcite and dolomite marble sequence prominently outcropping over most of Iraklia, forming a single large-scale tectonic window at the central and southern part of Iraklia (Figures 2 and 3b).

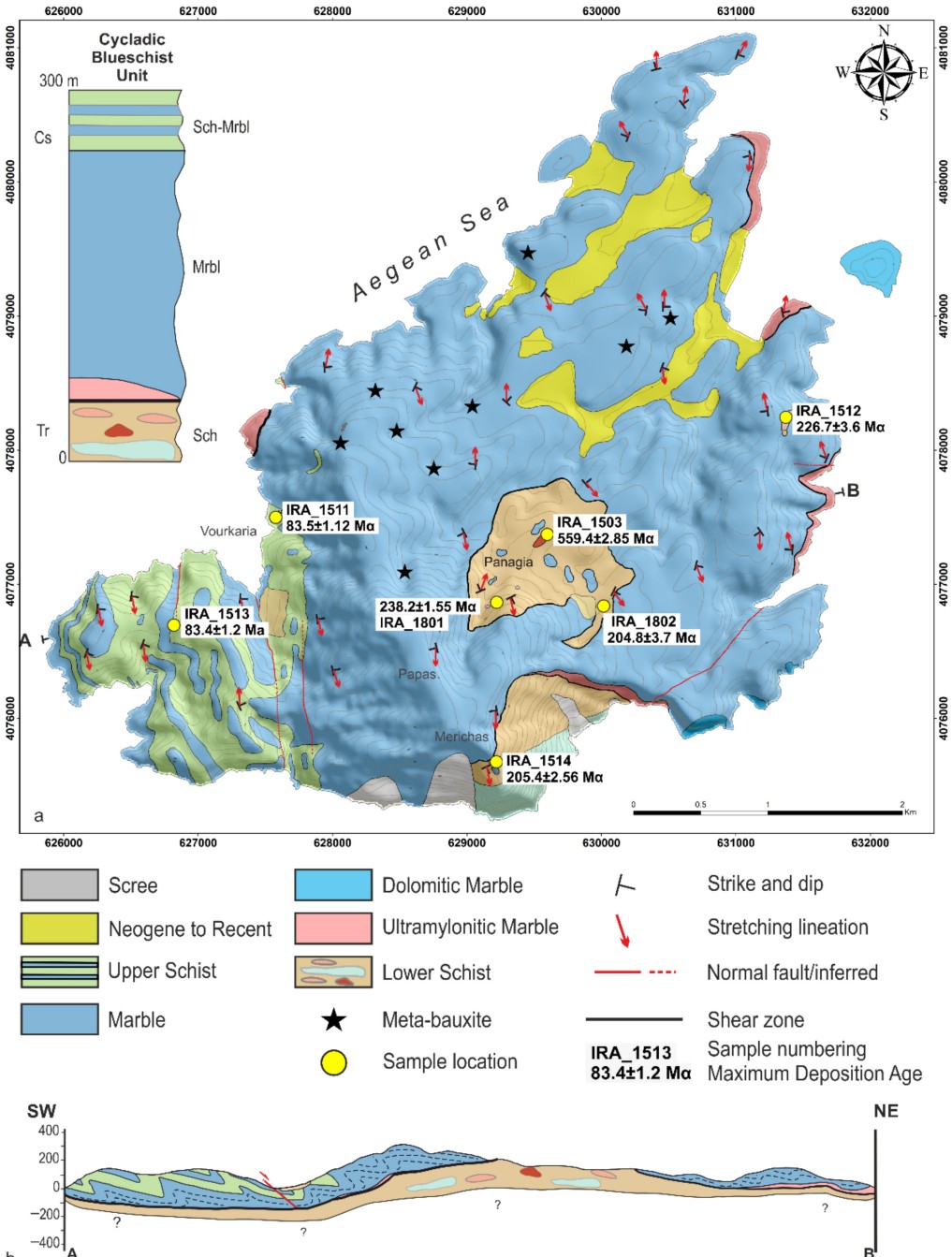

**Figure 2.** (**a**). Detailed geological map of Iraklia Island with metabauxite occurrences ([17], this study) and locations of samples collected for U-Pb detrital zircon dating (yellow circles) and (**b**) cross-section A–B.

The Lower Schist Fm. is juxtaposed structurally against the Marble Fm. along a ductile to brittle-ductile shear zone marked by the presence of a calcitic, fine-grained ultramylonitic marble at its base. This ultramylonitic marble is exposed only in a few places along the coast (Supplementary Figure S1). At the structural top of this Marble Fm., minor meta-bauxite occurrences are observed in north and north-western Iraklia [17,87] in this study.

In contrast, the contact between the Marble and Upper Schist Fm. appears to be gradational (depositional) and these two lithologies are folded together in large-scale N-S isoclinal folds.

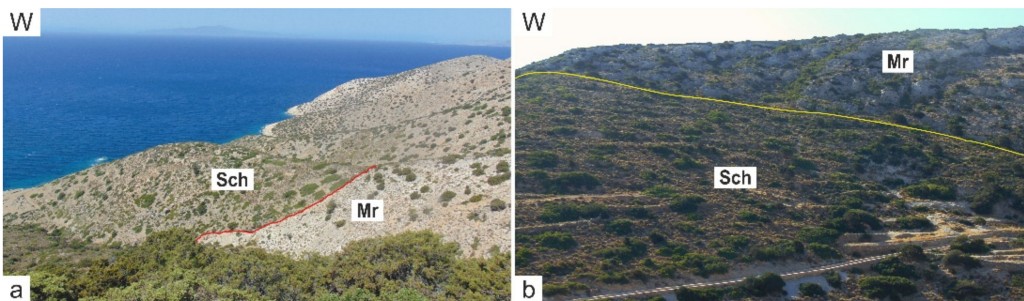

**Figure 3.** (**a**). View of the Upper Schist Fm. overlying the Marble Fm. (red line) in the SW part of the island (36°50′11″ N, 25°26′14″ E), (**b**). Main tectonic contact (yellow line) between the Lower Schist and the Marble Fm. from the central part of the island (36°49′57″ N, 25°27′2″ E).

*3.2. Structures*

Structural analysis on the CBU rocks on Iraklia revealed a three-stage deformation event (D1-D3), evolving from ductile to brittle conditions.

### 3.2.1. D1 Syn-Blueschist Facies Structures

The oldest D1 structure observed in thin sections represents a relic foliation S1 formed under blueschist-facies conditions, as revealed by the presence of blue amphibole. The HP-amphibole is preserved as inclusions within albite porphyroblasts that are syn-tectonic to the greenschist facies assemblages. In general, the albite is full of inclusions, usually forming an internal foliation (S1), sub-parallel to the main external S2 foliation. The growth of chlorite in between the boudinaged amphibole fragments strengthens the fact that the S2 foliation has been developed under greenschist-facies conditions (Figure 4a,b). It should be noted that the blueschist minerals are observed mainly in the Upper Schist Fm. in southwest Iraklia.

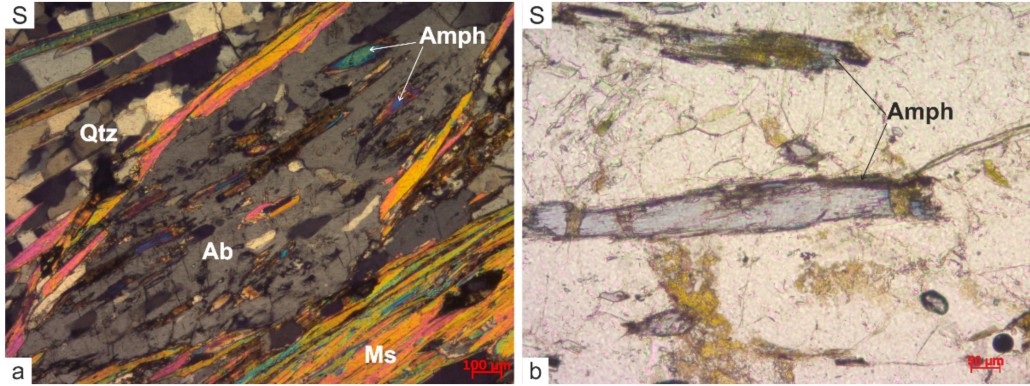

**Figure 4.** Microphotographs of glaucophane from the Upper Schist Fm. of SW Iraklia. (**a**). Albite porphyroblast with blue amphibole inclusions (crossed polars), and (**b**). elongated blue-amphibole grains boudinaged during the main D2 deformation stage. Note the chlorite grown in the interboudin area.

### 3.2.2. D2 Ductile to Brittle Deformation

The main D2 syn-metamorphic deformation stage is associated with the development of a dominant penetrative medium to fine-grained proto- to meso-mylonitic foliation (S2) that dips shallowly towards the east and west (Figure 5a). An N-S stretching lineation (L2), parallel to isoclinal fold axes (B2), is formed on the S2 foliation planes, defined by the aligned and stretched calcite-quartz-micas aggregates (Figure 5b,c). In thin sections, S2 foliation is formed by chlorite, epidote, actinolite, quartz, albite, white mica, and subordinate small garnet, attesting to greenschists-facies conditions. Dynamic recrystallization produced polygonal and non-equigranular quartz grains, which often exhibit subgrains. Undulose extinction of larger quartz grains indicates crystal–plastic deformation. Lensoid albite porphyroclasts bounded by chlorite and micas define a mylonitic foliation. Albite shows

intense brittle deformation during D2 shearing, but despite the fracturing, it preserves D1 blue amphibole. In addition, small syn- and post-tectonic garnet grains are observed. Paragneissic rocks are mainly composed of quartz, albite, chlorite, actinolite, minor micas, calcite and garnet; the latter represents pre-tectonic porphyroblasts developed during the HP event. Kinematically, ubiquitous S-C′ fabrics in most samples exhibit both top-to-the-N and S shearing, indicative of pure-shear deformation (Figure 6a–c). Similarly, symmetric albite clasts are frequently observed, suggestive of a strong coaxial component. In stark contrast, the calcite-rich ultramylonitic marbles record high-strain simple shear with the development of delta-clasts showing unambiguous top-to-the-N kinematics (Figure 6d).

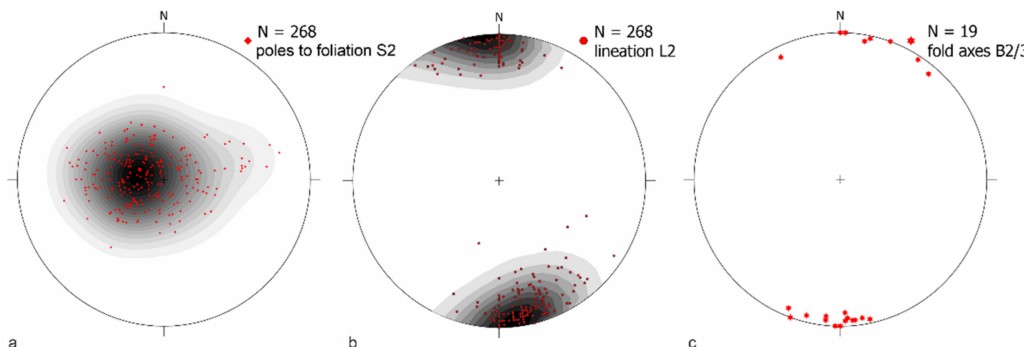

**Figure 5.** Stereoplots (lower hemisphere equal-area projection) (**a**). Poles to main foliation S2, (**b**). Stretching lineation L2 and (**c**) fold axes B2/3.

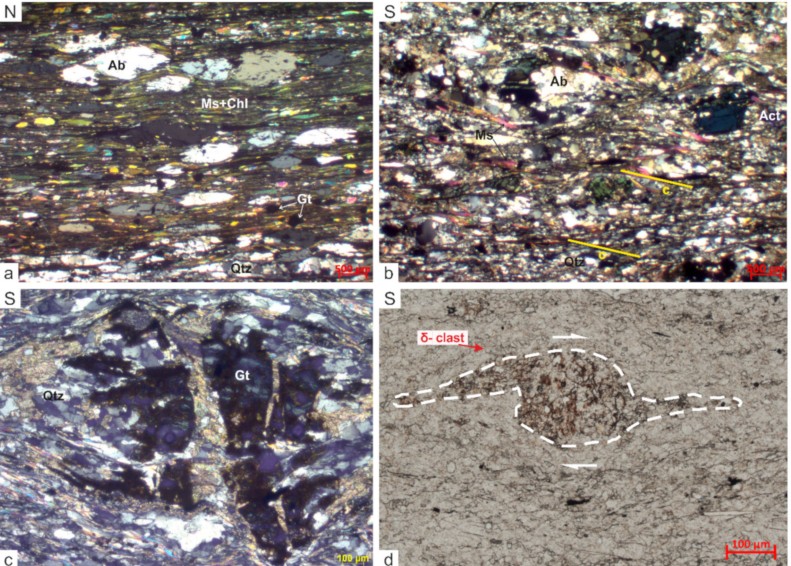

**Figure 6.** Thin sections (crossed polars) normal to the foliation and parallel to the stretching lineation (L2). (**a**) Upper Schist Fm. from the southwestern Iraklia (IRA_1509) comprising albite, quartz, chlorite, epidote, muscovite and rare post-tectonic garnet. S/C′ shear bands indicate top-to-the-north shearing, (**b**) top-to-the-N S/C′ shear bands in the paragneissic rock of the Lower Schist Fm. (IRA_1507), (**c**) boudinaged and rotated pre-tectonic garnet towards the S, and (**d**) Delta-type calcite+siderite clast indicating top-to-the-N shearing (white arrows).

Similar simple-shear top-to-the-N kinematics are also observed in the marble mylonites along the contact between the Lower Schist Fm. and the overlying Marble Fm. defining a mylonitic shear. It should be noted that near this contact, a younger brittle overprint is observed, producing an up to 1.5 m thick cataclasitic fault zone (Figure 7a). Progressively, the main S2 foliation is folded about N-S isoclinal folds and locally crenulated. An axial planar S3 cleavage is developed, associated with an N–S intersection lineation

(L3) parallel to the main N–S fold axis (B2/3) observed in the southern part of the island (Figures 5c and 7b).

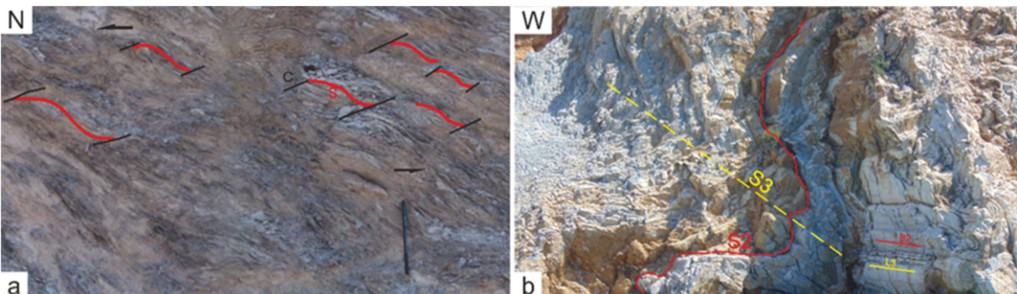

**Figure 7.** (**a**). S-C′ shear bands, developed in the Lower Schist Fm. with top-to-the-north kinematics (36°49′57″ N, 25°27′5″ E) and (**b**). Axial planar cleavage (S3) in the N–S isoclinal fold and an N–S intersection/stretching lineation L3 parallel to the B2/3 fold axes (southern part of the island) (36°49′13″ N, 25°27′4″ E).

### 3.2.3. D3 Late Brittle Deformation

D3 structures are related to brittle high-angle normal faulting that cut through all previous structures. The most significant faults are the NW-trending Vourkaria and the NE-trending Merihas fault zones (Figure 2). Kinematic indicators on the fault planes and the characteristic normal drag adjacent to the fault zones indicate a normal sense of slip.

## 4. Detrital Zircon U-Pb Geochronology and Results

### 4.1. Methodology

In this study, we performed detrital zircon (DZ) U-Pb dating on seven schist and paragneiss samples to constrain their provenance signature and Maximum Depositional Ages (MDAs) in order to provide insights into the pre-subduction evolution, tectonic affinity and regional tectonostratigraphic correlation of the metasedimentary CBU rocks of Iraklia.

Zircons were separated using standard separation techniques, including crushing and pulverizing, water table concentration, magnetic separation, and Bromoform and MEI heavy liquid separations at the UT Chron Laboratory at the University of Texas at Austin. Zircon grains were mounted on double-sided adhesive tape on 1-inch polycarbonate rounds. About 130 unpolished zircon grains were randomly selected from each sample for depth-profile Laser Ablation-Inductively Coupled Plasma-Mass Spectrometry (LA-ICP-MS) analysis, using a PhotonMachine Analyte G.2 excimer laser with a large-volume Helex cell connected to a Thermo Element2 ICP-MS, following the procedure outlined in [88] (Supplementary Figure S2). GJ1 was used as primary standard (601.7 ± 1.3 Ma; [89]) and Plešovice as secondary standard (337.1 ± 0.4 Ma; [90]). The VisualAge Data Reduction Scheme was used in Iolite v3.7 for correction of the U-Pb zircon data [91,92]. U-Pb ages of individual grains are reported with 2σ uncertainties. For dates >850 Ma, the $^{207}Pb/^{206}Pb$ date is reported, and for U-Pb dates <850 Ma, the $^{206}Pb/^{238}U$ dates are taken for the reported U-Pb age. A discordance filter 15% $^{206}Pb/^{238}U$ vs. $^{207}Pb/^{235}U$ and error filter 10% was applied to $^{206}Pb/^{238}U$ ages. A discordance filter 30% $^{206}Pb/^{238}U$ vs. $^{207}Pb/^{206}Pb$ was applied to $^{207}Pb/^{206}Pb$ ages. We used the detritalPy Python script [93] for data visualization and plotting of sample data as Kernel Density Estimates (KDEs). The analytical data and calculated ages of all Iraklia samples are listed in Table S1 of the Supplementary Material.

### 4.2. U-Pb Results

Five samples were collected from the Lower Schist Fm., of which three were from the quartz-mica schist (IRA_1801, IRA_1802, IRA_1503) and two from the paragneisses (IRA_1514, IRA_1512) (Figure 8, Supplementary Figure S3a–c). The quartz-rich schist IRA_1503 (*n* = 108) from the central part of the island close to Panagia village shows the

youngest zircon U-Pb age mode at 559.4 ± 2.8 Ma (Ediacaran). This sample contains only Precambrian (100%) zircons with a prominent Neoproterozoic (Tonian) peak at ~0.85 Ga, and a minor peak at the Paleoproterozoic/Archean boundary (~2.5 Ga). At the foot of Papas hill, the quartz-mica schist IRA_1801 (*n* = 130) reveals the youngest DZ age group at 238.2 ± 1.5 Ma (Carnian), with only a single zircon grain age at 220.1 ± 3.3 Ma, and zircon populations of Precambrian (*n* = 73, 52.6 %), Paleozoic (*n* = 50, 48%) and Mesozoic (*n* = 7, 5.4%) in age. The group of Paleozoic DZ ages shows a prominent Ordovician peak at 458.3 ± 6.5 Ma. Sample IRA_1802 (*n* = 118) reveals a similar age spectrum to sample IRA_1801 described above with DZ distributions of Precambrian (*n* = 54, 46%), Paleozoic (*n* = 42, 36%) and Mesozoic (*n* = 21, 18%) in age. However, for sample IRA_1802, the youngest DZ age component present is ~204.8 ± 2.9 Ma (Norian), with single zircon grain at 201.1 ± 5.2 Ma. Two prominent peaks are observed at ~243.2 ± 3.7 Ma (Anisian) and ~435.6 ± 7.4 Ma (Silurian). Depth-profile analysis from the two samples (IRA_1801, IRA_1802) from the Lower Schist Fm. revealed thin early Eocene metamorphic rims at about 48–49 Ma. Two zircons from each sample also yielded late Cretaceous metamorphic rims that experienced lead loss and hence were not included in the metamorphic age.

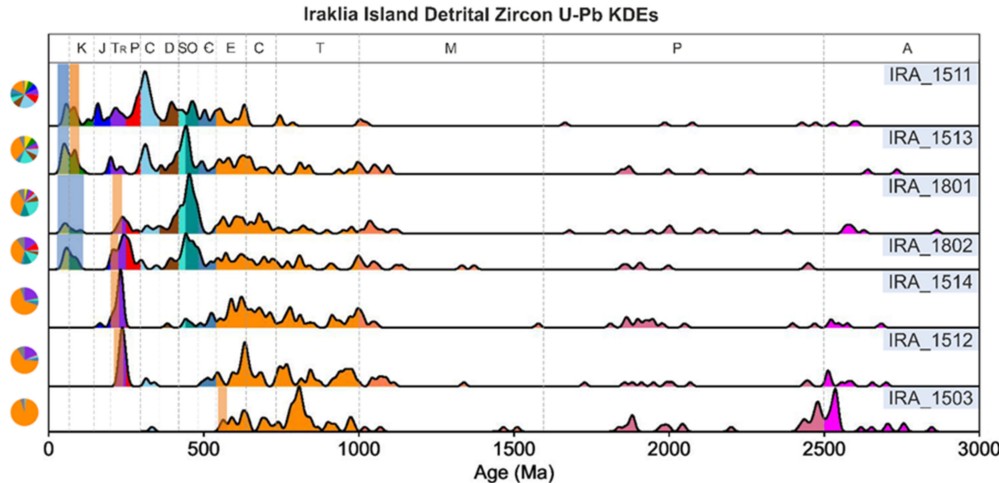

**Figure 8.** Detrital zircon U-Pb KDEs results. The diagrams show the compiled data from 3000-0 Ma for each sample. MDAs and metamorphic zircon rim ages are depicted in light-orange and blue, respectively. The pie diagrams show the proportions of core-age components.

Additionally, two paragneiss samples from the Lower Schist Fm. were analysed. IRA_1514 (*n* = 117) from the southern part of the island (Merihas) exhibits the youngest age mode at ~205.4 ± 2.6 Ma (Norian) with a single youngest zircon grain at 165.4 ± 1.8 Ma. It is also contained Precambrian (*n* = 88, 75%), Paleozoic (*n* = 10, 8%), and Mesozoic (*n* = 20, 17%) DZ age modes without any prominent peak, except for a dominant Triassic component at ~233.8 ± 2.7 Ma. In sample IRA_1512 (*n* = 120), Triassic zircon ages of ~226.7 ± 3.6 Ma (Carnian) also make up the youngest age mode of this sample from the north-eastern part of the island. Precambrian zircons (*n* = 94, 78%) make up the majority of the older detrital age modes in this sample, while it possesses small amounts of lower Paleozoic (*n* = 9, 7%) and Mesozoic (*n* = 18, 15%) DZ age components. A prominent Triassic peak at 231.4 ± 1.1 Ma and a minor peak at the Ediacaran/Cryogenian boundary are also present. Despite the similar Triassic peaks in both samples (IRA_1514 and IRA_1512), these two paragneisses show marked differences in the Paleozoic DZ age spectra. More specifically, sample IRA_1514 contains a small group of early Paleozoic (Silurian, Ordovician, and Devonian) DZ age modes, not observed in the sample IRA_1512, which is characterized by late Paleozoic (Carboniferous-Permian) DZ age components.

From the Upper Schist Fm. two quartz-mica schists were collected from the south-western and western parts of the island (Supplementary Figure S3d). Sample IRA_1513 (*n* = 105) shows the youngest DZ age mode at 83.4 ± 1.2 Ma (Campanian) with only one

zircon grain at 76.1 ± 2.8 Ma, and a DZ spectrum composed of Precambrian (*n* = 49, 47%), Paleozoic (*n* = 42, 40%) and Mesozoic (*n* = 14, 13%) ages. A major peak is present at the Ordovician/Silurian boundary, while a minor late Carboniferous peak is observed at ~315.2 ± 6.1 Ma (Pennsylvanian). Sample IRA_1511 (*n* = 134) exhibits a similar DZ age mode at ~83.5 ± 1.1 Ma (Campanian) with two zircon grains at 71.4 ± 2.5 and 72.8 ± 1.1 Ma and containing DZ spectra characterized by Precambrian (*n* = 31, 23%), Paleozoic (*n* = 75, 56%) and Mesozoic (*n* = 28, 21%) DZ ages. Both samples also yielded thin early Cenozoic metamorphic rims with ages of 49–56 Ma (Early Eocene), similar to the rims obtained from the Lower Schist Fm. (Figure 8, Supplementary Figure S4).

The Th/U in zircon grains from all Iraklia samples are useful to elucidate their origin. Most detrital zircons from the Lower and Upper Schist Fm. were derived from a felsic origin, as indicated by their elevated Th/U (>0.1, Figure 9). In contrast, only a minor component of zircons shows metamorphic (Th/U < 0.1; [94]) and mafic-derived sources (Th/U > 1.5, [95]). In addition, a small group of zircons is found in the transition zone between the felsic and mafic origin (1 < Th/U < 1.5, [95,96]).

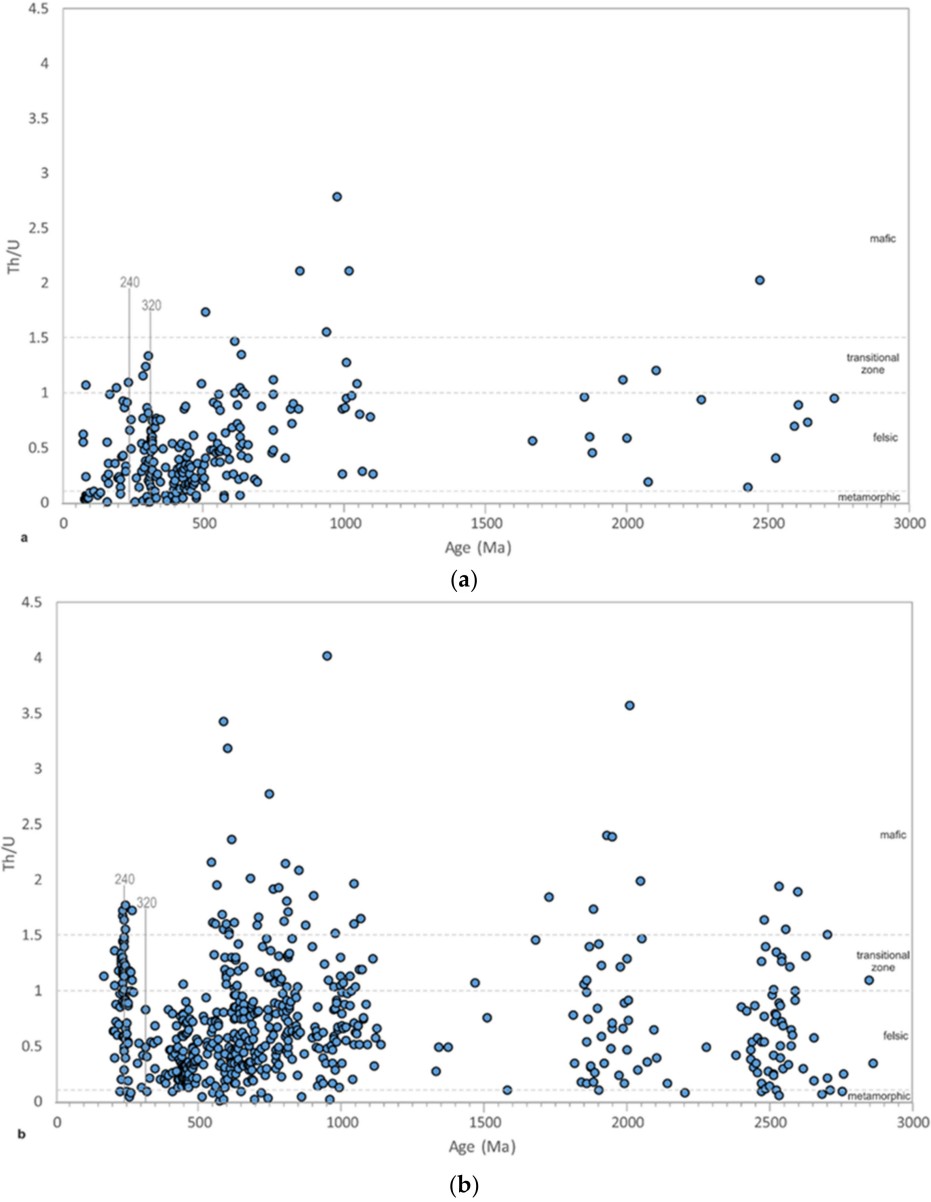

**Figure 9.** Zircon Th/U vs. age diagram from (**a**) the Upper Schist Fm. and (**b**) the Lower Schist Fm. of Iraklia island.

## 5. Discussion

### 5.1. Review of the Lithostratigraphy of Iraklia

The CBU rocks on Iraklia Island have been previously considered a Triassic sequence consisting of schist–marble alternations and an overlying marble [16,17]. However, our new DZ U-Pb data allow us to determine the MDAs and investigate the sedimentary provenance and, consequently, the tectonic affinity of the different rock units on Iraklia. Following the described method [97], the youngest age modes (3+ grains) that overlap within 2-sigma error, excluding metamorphic rims, were used to calculate MDA in order to establish a new chronostratigraphic and refined lithostratigraphic framework for the Iraklia CBU section. On the basis of these data, the schists from the central and southwestern part of the island can be differentiated based on their MDAs, provenance, and lithology, as well as their structural position with respect to the intermediate Marble Fm.

The heterogeneous Lower Schist Fm. consists of various lithologies, including quartz-mica schists with lenses of paragneiss rocks and quartz-rich schist. The quartz-mica schist yielded late Triassic MDAs (Carnian-Rhaetian) ranging from 238.2 ± 1.5 to 204.8 ± 2.9 Ma. The paragneisses also gave late Triassic MDAs (Norian-Rhaetian) with ages of 226.7 ± 3.6 and 205.4 ± 2.6 Ma, while the quartz-rich schist yielded a Neoproterozoic (Ediacaran) MDA at about 559.4 ± 2.8 Ma. The contact with the overlying Marble Fm. is a ductile to brittle-ductile shear zone. Not surprisingly, given the highly strained marbles, no fossils have been reported from the carbonate rocks, implying that the obtained Carnian-Rhaetian MDAs for the underlying Lower Schist Fm. should be considered the upper age limit for the depositional age. The major age mode in the Lower Schist Fm. points to strong Triassic syn-rift, bimodal volcanic rocks and is indicative of a Triassic age for the Lower Schist Fm.

The Upper Schist Fm. mainly includes quartz-mica schists and surprisingly yielded late Cretaceous MDAs of 83.5 ± 1.12 and 83.4 ± 1.2 Ma (Campanian). The late Cretaceous MDAs and the early Eocene (Ypresian) U-Pb metamorphic rims bracket the true deposition age of the metasediments composing the Upper Schist Fm. as latest Cretaceous to Paleocene. These schists are isoclinally folded with the underlying Marble Fm. Despite evidence for gradational stratigraphic contact, observed in a few places, an isoclinally folded early, prograde tectonic contact cannot be excluded. Interestingly, the metabauxite occurrences within the Marble Fm. in north-western Iraklia reflect a short emersion period and characterize shallow-water platform facies. Similar metabauxite lenses have also been observed on north-eastern Naxos [35] and Amorgos island [98,99]. The occurrences of Amorgos were correlated with the Cretaceous bauxites of continental Greece [100]. Given the latest Cretaceous to Paleocene age of the overlying Upper Schist Fm., it seems plausible to posit a Cretaceous age for these metabauxites and the enveloping Marble Fm.

### 5.2. Structures and Metamorphism

Structural analysis on Iraklia island shows successive deformation events related mainly to syn-convergent prograde and back-arc extensional exhumation processes. Macro- and microstructures associated with the prograde path and the peak HP metamorphism are almost entirely obliterated.

This earlier HP event is only preserved as relic blueschist facies minerals, such as blue amphiboles and probably garnet in rocks of the Upper Schist Fm. and the paragneissic rocks, respectively. Additionally, strong evidence for the HP metamorphism is supported by thin, low Th/U zircon rims with U-Pb ages of ~48–56 Ma from both the Lower and Upper Schist Fm. These early Eocene metamorphic ages are in broad agreement with ages from HP/LT rocks in several Cycladic islands clustering between 55–40 Ma [11,42,101–106]. Isotopic dating on Ios and SE Naxos islands also revealed Eocene HP-LT metamorphism at ~55–40 Ma [15,107,108] and 40 Ma, respectively [55,109], suggesting a prolonged or multi-stage HP-LT event in the Cyclades. Importantly, however, the Eocene zircon rims attest that the Iraklia rocks are part of the same CBU HP-LT metamorphic assemblage documented on the CBU rocks regionally.

The main D2 deformation event produced N-S isoclinal folding with a penetrative S2 axial planar foliation and accompanied by the mineral/stretching lineation (L2). Based on the mineral assemblage, this metamorphic overprint of the Iraklia CBU rocks probably did not exceed the greenschist facies P-T conditions. In contrast, Naxos island was affected by high-grade post-HP-LT Barrovian-type metamorphism [26,41,110–114].

Shear deformation was localized along the tectonic contact between the Lower Schist Fm. and the overlying Marble Fm. Ubiquitous shear bands along this contact display consistent top-to-the-north kinematics and advocate for internal deformation within lower-plate CBU rocks on Iraklia island. Given the kinematic similarities with extensional fabrics on Naxos, this intra-CBU contact likely represents a shear zone associated with exhumation and extensional crustal attenuation. This shear zone appears to account for the ductile lower-plate thinning of the CBU column in Iraklia. Similar ductile-brittle shear zones have been described within CBU from several Cycladic islands, all related to extensional shearing [58,81,115].

*5.3. Provenance of Detrital Zircons*

The rocks of the CBU on Iraklia can be subdivided into Triassic and late Cretaceous lithostratigraphic packages on the basis of their MDA constraints. The DZ age spectra of the Lower and Upper Schist Fm. with their distinct ages thus record the sedimentary provenance evolution of two fundamentally different geodynamic environments. Hence, the DZ provenance evolution and changes have the potential to shed light on tectonic and palinspastic configuration during both late Triassic rift margin evolution associated with opening of the Neotethys and late Cretaceous–Paleocene convergence and subduction of the Neotethys.

In the Lower Schist Fm., the DZ age spectra include a prominent signal of Early Paleozoic ages, a significant contribution of Neoproterozoic (Pan-African) ages, and a minor Late Paleozoic input. The combination of Neoproterozoic and Lower Paleozoic DZ input likely reflects eroded material from peri-Gondwanan terranes and points to uplift and erosion of the pre-Variscan crystalline basement, which served regionally as a source terrane. It should be noted that the abundant Neoproterozoic input might represent first-cycle basement derivation. However, it could just as likely be derived from the recycling of early Paleozoic Gondwanan strata in light of several cycles of the Gondwanan basement erosion, deposition, and recycling. Prominent early Paleozoic–Neoproterozoic ages have also ascertained from pre-Variscan CB rocks in the southern Cyclades [14,15,20] and therefore likely represent a local source, especially during Permian-Triassic rifting. This is also supported by early exhumation of basement rocks described from both the southern Cyclades (e.g., Ios, [14]) and the Pelagonian domain [116]. This is also reinforced by the results obtained from the quartz-rich schist layer within the Lower Schist Fm. However, the nearly unimodal Pan-African (Ediacaran) DZ age spectrum of the quartz-rich schist suggests either erosion of juvenile felsic crust from the East African orogeny or recycling of Late Neoproterozoic quartzites [117,118]. The high amount of Tonian DZ ages with a pronounced peak at 850 Ma points to the Minoan terranes, characteristic of the basement rocks exposed on the External Hellenides [119–123]. On Ios island, Neoproterozoic metasedimentary sequences yielded similar MDAs and dominant Pan-African provenance and represent slivers of the pre-Carboniferous metasedimentary rocks of the CB intruded by voluminous Carboniferous granitoids [14]. The minor late Paleozoic DZ input of the Lower Schist Fm. likely reflects eroded upper-crustal plutonic or volcano-sedimentary material associated with the active Eurasian convergent margin linked to Paleotethys closure. Late Carboniferous–early Permian magmatism characterizes the basement rocks in the Cyclades [14,15,45,75,118,119,124,125], the external Hellenides and Pelagonian domain [116,119,122,126,127].

Major Triassic DZ age input, associated with the Triassic magmatic activity, is significant in the paragneissic rocks and the quartz-mica schist of the Lower Schist Fm. The prominent Ladinian–Carnian peaks in the paragneisses reflect the long-lived bimodal syn-

rift magmatic activity during Neotethyan opening. This is directly supported by the new Th/U data for the ~240 Ma Triassic detrital zircons, suggesting they originated from rocks with felsic and intermediate composition and a small number of mafic-derived zircons, attesting to bimodal syn-rift magmatism and opening of the Neotethyan oceanic basin (e.g., Pindos basin, southern Neotethys, [122,123,128–131]). It is also striking that the DZ data from the Lower Schist Fm. exhibit remarkable similarities in terms of MDAs and provenance with the described Permian to Triassic CBU metasedimentary on Ios and Sikinos islands [15], implying a close spatial and genetic relationship.

The Upper Schist Fm. on Iraklia is characterized by late Cretaceous MDAs and exhibits DZ age spectra that show diminished Neoproterozoic, still significant early Paleozoic, and dominant late Paleozoic and Mesozoic DZ U-Pb age components. The abundant late Paleozoic–Mesozoic age modes probably reflect the input of recycled crystalline basement and metasedimentary rocks from the Pelagonian domain and/or more internal Hellenide domains as source rocks. Interestingly, the Th/U data for most Cretaceous detrital zircons point to a metamorphic origin with a limited contribution of zircons from felsic protoliths. These schists show similarities in provenance with the other late Cretaceous CBU schists from the Western Cyclades [25], northern Ios and Sikinos [15], and/or to the latest Cretaceous passive-margin sequence of Naxos island [52,54,132]. This pronounced provenance shift suggests that the late Cretaceous Upper Schist Fm. of the CBU, as well as other similar late Cretaceous CBU rocks, received sediments from the upper-plate of the convergent margin and likely represents trench fill deposited shortly before subduction.

### 5.4. Regional Implications

Our provenance- and MDA-based correlation between age-equivalent CBU rocks of Iraklia and neighboring islands provides new insights into the paleogeography before the Cenozoic subduction and the related tectonometamorphic evolution. These data also illuminate the pre-subduction history of the island from the middle–late Triassic to late Cretaceous (Figure 10) and establish a new stratigraphic column subdivided into two major diachronous packages.

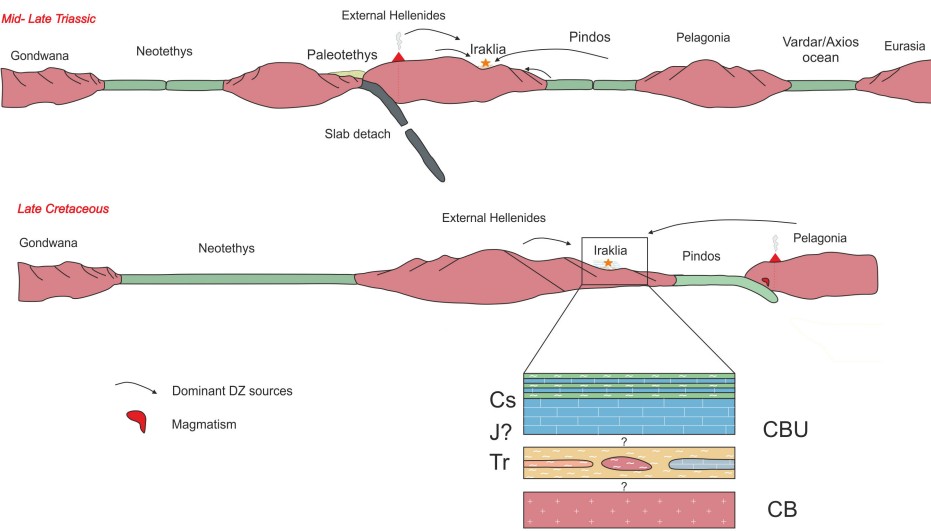

**Figure 10.** Schematic paleogeographic reconstruction of Iraklia island from middle–late Triassic to late Cretaceous time.

Our new data demonstrate that the lower portion CBU rocks in Iraklia were initially deposited in a locally sourced, isolated late Triassic syn-rift basin, receiving mainly recycled Gondwanan-affinity detrital zircons. This indicates that older, pre-Variscan basement rocks underwent uplift and erosion, supplying recycled sediment to the Lower Schist Fm. During this time, deposition was also accompanied by locally bimodal syn-rift volcanism, as evidenced by abundant Triassic zircons with felsic, intermediate, and mafic Th-U sig-

natures. Overall Permian to Triassic magmatism was likely associated with both the final closure of southern Paleotethys that took place in Ladinian–early Carnian and early rifting and opening of Neotethyan basins [14,15,48,75,122–124,128–131,133,134]. This late Triassic Lower Schist Fm. contains abundant Neoproterozoic Pan-African DZ ages, implying that sedimentation occurred in a proximal position to Ios island. This is also supported by the similarity of late Triassic metasedimentary rocks observed on both islands and the notion that these rocks are laterally equivalent lithological units, although sedimentation in northern Ios and Sikinos probably started in Permian times [15]. However, no rocks structurally or stratigraphically lower than the Triassic Lower Schist Fm. are exposed in Iraklia. This lithostratigraphic correlation is also supported by the quartz-rich schist, characterized exclusively by unimodal Neoproterozoic DZ U-Pb ages that were probably sourced from pre-Variscan metasedimentary CB rocks on Ios. This also strengthens arguments for an early syn-rift unroofing and exhumation of crystalline basement rocks in a broader area. Sedimentation continued in a passive margin setting with deposition of shallow-water carbonates probably of (Jurassic?–)Cretaceous age, interrupted by short emersion episodes and bauxite deposits.

The Upper Schist Fm. on Iraklia, represents the final and fundamentally different episode of the depositional history of the CBU. It is marked by late Cretaceous quartz-mica schists that correlate well with CBU rocks in northern Ios [15] and Naxos [52,132] islands, exhibiting a pronounced shift in DZ provenance dominated by Mesozoic and late Paleozoic and a dramatic reduction in Neoproterozoic age components. Sediments were likely derived from rocks exposed in the internal Hellenides, including the Pelagonian domain and Rhodope belt, and likely correspond to clastic sedimentation in the subduction trench with sediment sourced from the overriding plate and delivered to the convergent margin by fluvial systems from the hinterland.

Relic blue amphibole and ages of metamorphic zircon rims obtained from both CBU schist Fms., show that all the Iraklia rocks experienced Early Eocene HP subduction metamorphism, overprinted by a younger greenschist-facies metamorphism similar to Ios island [79,80,135]. However, they have not experienced the high-grade metamorphic conditions observed in the Naxos MCC associated with migmatitic dome formation [110–112]. Kinematic indicators demonstrate dominant top-to-the-N shearing during the main greenschist-facies deformation event. Top-to-the-S kinematics were also observed locally, but the lack of cross-cutting relationships does not allow a temporal differentiation between the different kinematics. However, it is also plausible that the opposing kinematics could be related to a general shear deformation pattern. Although no geochronological data could constrain the age of this intra-Cycladic Blueschists Unit shear zone of Iraklia, it presents kinematic similarities with the P-NDS. Similarly, the top-to-the-S kinematics is compatible with the South Cycladic Shear Zone [61,63,80] and/or the Santorini Detachment System [13] active concomitantly with the P-NDS. Although it is tempting to correlate the dominant top-to-the-N and subordinate top-to-the S-kinematics of Iraklia with the large-scale detachment faults, the latter juxtaposes either the Pelagonian-derived units against the CBU (P-NDS, SDS) or the CBU against the CB (SCSZ). Nonetheless, the intra-Cycladic Blueschists Unit shear zone of Iraklia has likely acted in a lower structural level, synchronous to the crustal-scale structures of the south Aegean domain, contributing to the attenuation of the CBU rocks probably in the footwall of P-NDS (Figure 11).

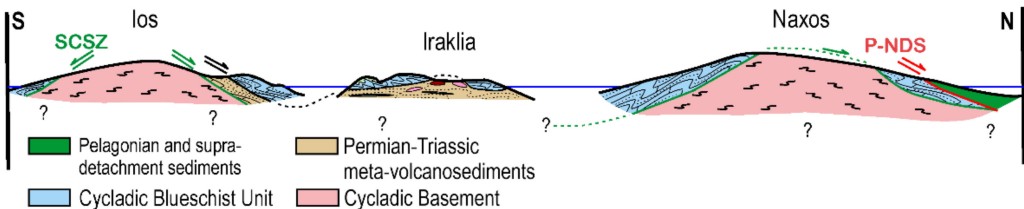

**Figure 11.** Simplified sketch illustrating the lithostratigraphic and structural correlations between Iraklia and the neighboring Ios and Naxos islands.

## 6. Conclusions

The island of Iraklia exposes a heterogeneous stratigraphic assemblage of CBU rocks. New DZ data show that this tectonostratigraphic column can be subdivided into distinct Lower and Upper Schist Fms., separated by a deformed marble horizon. The Lower Schist represents a middle to late Triassic meta-clastic sequence that likely represents a locally sourced, syn-rift sequence that also received Triassic bimodal volcanic input. DZ U-Pb age spectra from the Lower Schist revealed a dominant Precambrian and a Lower Paleozoic signature supporting local sourcing and are similar to a pre-Variscan basement exposed on Ios.

The Lower Schist Fm. of the Iraklia CBU is structurally juxtaposed against an overlying Marble Fm. with bauxite lenses that exhibit abundant top-to-the-N ductile shear zone fabrics. Based on the regionally correlated occurrence of bauxite, the stratigraphic age of this formation is likely late Cretaceous—a notion supported by the MDAs from the overlying Upper Schist Fm. This age also suggests a significant intra-CBU depositional hiatus or structural omission—an omission would be compatible with the top-to-the-N shear zone that appears to have structurally attenuated the CBU in the footwall of the P-NDS.

The Upper Schist Fm. represents a late Cretaceous meta-clastic sequence with a distinctly different provenance signature, dominated by late Paleozoic and Mesozoic DZ U-Pb ages. This signals for a prominent sediment input from the internal Hellenides, including the Pelagonian domain, and appears to correspond to clastic sedimentation in the subduction trench with sediment sourced from the overriding plate prior to early Paleogene subduction, underplating, and metamorphism.

The zircon rim U-Pb ages from both CBU clastic formations on Iraklia revealed early Eocene HP-LT subduction metamorphism, similar to events documented in the neighboring islands of Ios, Sikinos, and Naxos, and is widespread throughout the Cyclades.

The Lower and Upper Schist Fms. of Iraklia bear significant similarities with Triassic and Cretaceous CBU schists, respectively, from the Ios and Sikinos islands, suggesting their close relationship during deposition and similar structural position during both Triassic rifting and during Cenozoic back-arc extension and detachment faulting in the attenuated footwall of the P-NDS.

**Supplementary Materials:** The following are available online at https://www.mdpi.com/article/10.3390/min12010083/s1, Table S1: U-Pb detrital zircon analyses from Iraklia Island, central Cyclades, Greece, Figure S1: (a) View of the contacts between the ultramylonitic marble and the underlying Lower Schists (yellow dashed line) and the overlying Marble Fm. (orange dashed line) (36°49′41.43″ N 25°27′40.13″ E), (b) S-trending stretching lineation from the ultramylonitic marble (36°49′22.70″ N 25°27′9.53″ E) and (c) Panoramic view of the contact between the ultramylonitic marble and the Marble Fm. (red dashed line), observed along the southern coastline of Iraklia (36°49′35.84″ N 25°28′29.85″ E), Figure S2: Mosaic photos of detrital zircon grains from Iraklia Island: (a) sample IRA_1512, (b) sample IRA_1802 and (c) sample IRA_1511, Figure S3: Representative photos of collected samples from Iraklia Island: (a) Precambrian quartz-mica schist (36°50′14.50″ N 25°27′22.24″ E), (b) Middle-Late Triassic quartz-mica schist (36°50′1.62″ N 25°27′42.06″ E), (c) Middle-Late Triassic paragneiss (36°49′22.71″ N 25°27′6.58″ E), and (d) Late Cretaceous quartz-mica schist (36°49′54.59″ N 25°25′51.07″ E) and Figure S4: Rim ages and MSWDs calculated by the weighted mean of $^{206}Pb/^{238}U$ ages for concordance. IsoplotR was used for calculations [136].

**Author Contributions:** S.L. (Sofia Laskari) project conception, investigation, lab analyses, writing—original draft preparation; K.S. project conception, investigation, editing; S.L. (Stylianos Lozios) investigation, editing; D.F.S., lab analyses investigation, editing; E.M.P., lab analyses investigation, editing and C.S. investigation, editing. All authors have read and agreed to the published version of the manuscript.

**Funding:** This research was funded internally by the National and Kapodistrian University of Athens and the Univerisity of Texas at Austin, including the UTChron laboratory.

**Data Availability Statement:** The data presented in this study are available in the Supplementary Materials.

**Acknowledgments:** The authors are grateful to Lisa Stockli, Daniel Arnost, and Spencer Seman for their invaluable assistance in the UTChron laboratory.

**Conflicts of Interest:** The authors declare no conflict of interest.

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
