# Peer review of "Structural Study and Detrital Zircon Provenance Analysis of the Cycladic Blueschist Unit Rocks from Iraklia Island: From the Paleozoic Basement Unroofing to the Cenozoic Exhumation"

_minerals, doi:10.3390/min12010083_

Round 1

Reviewer 1 Report

Title: Provenance analysis and structural study of the Cycladic Blueschist Unit rocks from Iraklia island: from the Paleozoic basement unroofing to the Cenozoic exhumation.
Authors: Sofia Laskari et al.
Journal: Minerals

This is an excellent paper, carefully documenting the field geology and U/Pb datings in a very poorly known area of the Cyclades that are conversely very well studied.

I annotated the pdf version with quite a lot of details. All my comments are then available on this copy! The only valuable and very general addition to it is the need to better emphazise the field data including the map, the structural results (by adding plots, stats... pictures) that may really add to the geology of Iraklia! Accordingly, i think that a section virtually cutting across Ios, Iraklia and Naxos would also be a good maner to valorize structural data in the discussion before the upscale that is currently proposed.

Author Response

We thank the reviewer for the detailed review of our manuscript, which significantly helped us improve it, especially regarding the clarity of the text.

We have accepted almost all suggestions, including:

  • the change of the title,
  • improving some figures and addition of figures with stereo plots (now figure 7), a cross-section from Ios to Naxos through Iraklia (figure 11)
  • points made about the crustal-scale structures in the south Aegean and correlation with the Iraklia shear zone that we have described

We also made clear in the text that we did not find the Cycladic Basement on Iraklia but a lense within the Triassic

Regarding the detrital zircons and the Th/U vs. age diagrams, we stick to our initial organization of the manuscript. These are first-order results that aid in clarifying the lithostratigraphy and the structural relationships.

Last but not least, we have corrected the typos that the reviewer noted.

We have uploaded a pdf file with replies to all comments made by the reviewer

Sofia Laskari

Reviewer 2 Report

Dear authors, the text I have read I like, it is written correctly, however I have some suggestions:
1. I will start with the title, since the authors discuss zircon research and rely on this in their arguments, it would be worthwhile to indicate this in the title. 
2. in fig 1 in the schematic map, wouldn't it be possible to include more details or put also the tectonic map of the Greek islands separately? It would make things much clearer for the reader.
The methodology is given in the text as subsection 4.1. I suggest moving it higher above the geological data and describing there also field work, microscopy, etc. 
4. results lack photographs of typical representatives of zircon populations. It would be worthwhile to show at least one example each of the discussed zircon types. After all, this is the most important part of this article.
In line 66 there is a question mark, maybe it means uncertainty but if so it would be better to put it in brackets (?).

Translated with www.DeepL.com/Translator (free version)

Author Response

We thank the reviewer for the good comments and the positive review

We have addressed all the issues pointed by the reviewer except for the cl images..

Our detailed answers are in the attached word file

Best regards

Reviewer 3 Report

Dear Authors and Editors,

I attach a file with a few minor comments. Practically, I accept the paper in its current form. Great Job!

Best,

Reviewer

Author Response

We thank the reviewer for the good comments. We have addressed almost all the issues pointed by the reviewer, except for the cl images, which we cannot add

Our responses are included in the word file attached
